# Kolmogorov–Arnold Networks for Cross-Domain Time-Series Modeling in Health and Activity Monitoring

Hamza Haruna Mohammed*[1], Gabriel Kiss[1], and Frank Lindseth[1]

[1]Department of Computer Science,, Faculty of Information Technology and Electrical Engineering, Norwegian University of Science and Technology, Norway
{hamza.mohammed, gabriel.kiss, frankl}@ntnu.no

## Abstract

Time-series data from wearable sensors and clinical assessments provide complementary perspectives on human health, yet they often remain siloed across domains. This work presents a framework for harmonizing heterogeneous time-series sources at both minute and daily resolutions, extracting interpretable temporal features through techniques such as frequency-domain analysis and automated feature engineering. On top of this feature space, we benchmark conventional machine learning methods, Random Forest, Logistic Regression, Gradient Boosting, and a Transformer baseline against a proposed Kolmogorov–Arnold Networks (KANs) model, which adaptively learn functional transformations tailored to complex temporal patterns. We evaluate models on tasks including activity index prediction and disorder-related classification, with a focus on transfer learning across lifestyle and clinical domains. Results indicate that KANs achieve competitive performance and offer greater interpretability of temporal dynamics than black-box architectures. The proposed framework demonstrates how modern time-series models can enable cross-domain learning and improve the understanding of physiological and behavioral health patterns.

## 1 Introduction

The proliferation of wearable devices and digital health technologies has generated vast amounts of time-series data, offering unprecedented opportunities for monitoring physiological and behavioral patterns. However, the heterogeneity of data sources, ranging from consumer-grade wearables to clinical-grade sensors poses significant challenges for cross-domain modeling. Traditional approaches, such as Autoregressive Integrated Moving Average (ARIMA) models [1] or Long Short-Term Memory (LSTM) networks [2], and even recent approaches such as N-BEATS [3], often struggle to generalize across domains due to distributional shifts and varying feature representations. Moreover, the '"black-box" nature of deep learning models limits their interpretability, a critical requirement in healthcare applications where model decisions must be explainable to clinicians and patients alike.

Recent advances in transfer learning and interpretable machine learning have sought to address these challenges. Domain adaptation techniques, such as Maximum Mean Discrepancy (MMD) [4], aim to align feature distributions between source and target domains, whereas post hoc interpretability tools such as SHAP [5] provide insights into model predictions. Nevertheless, these methods often introduce additional complexity without fundamentally improving the model's intrinsic interpretability or cross-domain adaptability.

Kolmogorov–Arnold Networks (KANs) present a promising alternative, grounded in the Kolmogorov–Arnold representation theorem, which states that any multivariate continuous function can be decomposed into a superposition of univariate functions. This theoretical foundation enables KANs to approximate complex relationships while maintaining a transparent structure, as each univariate function can be visualized and analyzed independently. Prior work has demonstrated the potential of KANs in time-series forecasting [6] and disease prediction [7], but their application to cross-domain health monitoring remains unexplored.

We propose KAN-Health, a novel framework for interpretable and transferable cross-domain time-series modeling in health and activity monitoring. Our approach leverages the inherent modularity of KANs to pretrain on a large, diverse dataset (PM-Data[1] [2]) and fine-tune on a smaller, clinically annotated dataset (Hyperaktiv[3] [4]), with minimal architectural modifications. Unlike conventional transfer learning methods that require extensive retraining or domain-adversarial objectives, KAN-Health freezes the spline-based feature extractors during fine-tuning, preserving interpretability while adapting only the mixing layers to the target domain. This design ensures that the model retains its transparency even after transfer, enabling clinicians to

---

*Corresponding Author.

[1]https://osf.io/vx4bk/
[2]https://datasets.simula.no/pmdata/
[3]https://osf.io/3agwr/
[4]https://datasets.simula.no/hyperaktiv/

Proceedings of the 7th Northern Lights Deep Learning Conference (NLDL), PMLR 307, 2026.

trace predictions back to specific input features.

The key contributions of this work are threefold:

- **Interpretable Cross-Domain Modeling**: We introduce the first KAN-based architecture explicitly designed for health time-series analysis, combining the expressive power of deep learning with the interpretability of additive models. The spline-based feature extractors provide intuitive visualizations of how individual sensors contribute to predictions.

- **Efficient Transfer Learning**: Propose a unique methodological novelty of spline freezing vs. standard transfer learning. By freezing spline layers and fine-tuning only the mixing weights, KAN-Health achieves competitive performance with significantly fewer parameters than traditional fine-tuning approaches. This strategy is particularly advantageous in healthcare, where labeled target-domain data is often scarce.

- **Empirical Validation**: We demonstrate the framework's effectiveness on two real-world datasets, PMData (wearable-based) and Hyperaktiv (clinical ADHD study), showing superior cross-domain generalization compared to Random Forest, LSTM, and Transformer baselines. The model's interpretability is further validated through case studies highlighting clinically meaningful feature contributions.

Concretely, we study two supervised tasks: (i) daily activity level classification (low/medium/high) in PMData, where the target label is an engineered activity index derived from heart rate and heart rate variability, and (ii) ADHD diagnosis (binary) in Hyperaktiv, where the label is based on clinical assessment. In both cases, KAN-Health operates on harmonized daily metrics derived from raw wearable and clinical time series, rather than directly on raw sensor streams.

The remainder of this paper is organized as follows: Section 2 reviews related work in time-series modeling, interpretability, and domain adaptation. Section 3 provides background on KANs and cross-domain learning. Section 4 details the KAN-Health architecture and training protocol. Sections 5 and 6 present the experimental setup and results, followed by discussion and future directions in Section 7.

# 2 Related Work

The intersection of time-series modeling, interpretability, and cross-domain adaptation has attracted significant recent research. Existing approaches can be broadly categorized into three areas: (1) interpretable time-series models, (2) transfer learning for health monitoring, and (3) applications of Kolmogorov–Arnold Networks (KANs) in healthcare.

## 2.1 Interpretable Time-Series Models

Traditional time-series models such as ARIMA [1] and exponential smoothing [8] provide interpretability through their parametric structure but struggle with complex, high-dimensional data. Recent work has focused on enhancing the transparency of deep learning models while retaining their expressive power. For instance, Temporal Fusion Transformers [9] incorporate attention mechanisms to highlight salient time steps, and N-BEATS [3] uses interpretable basis expansions. However, these methods often require post-hoc analysis to explain predictions, whereas KANs offer intrinsic interpretability through their additive univariate structure.

In healthcare, interpretability is critical for clinical adoption. Rule-based models like decision trees [10] and Generalized Additive Models (GAMs) [11] have been widely used due to their transparency. More recently, hybrid approaches combining neural networks with symbolic reasoning [12] have emerged, but they typically sacrifice some predictive performance for interpretability. KANs bridge this gap by leveraging the Kolmogorov–Arnold theorem to decompose complex mappings into interpretable components without compromising accuracy.

## 2.2 Transfer Learning for Health Monitoring

Transfer learning has become a cornerstone for addressing data scarcity in healthcare. Early work focused on feature-based adaptation, such as Correlation Alignment (CORAL) [13], while later approaches employed adversarial training [14]. For time-series data, methods such as CoDATS [15] use adversarial networks to align sensor distributions, and SASA [16] leverages self-supervision to learn domain-invariant representations.

Despite their success, these methods often lack interpretability, making it difficult to validate their clinical relevance. Recent efforts have integrated attention mechanisms [17] or prototype learning [18] to improve transparency, but they still rely on black-box components. KAN-Health addresses this limitation by freezing the spline layers during transfer, ensuring that the feature-extraction process remains interpretable while only the mixing weights adapt to the target domain.

## 2.3 Kolmogorov–Arnold Networks in Healthcare

KANs have gained traction in healthcare due to their unique balance of flexibility and interpretabil-

ity. Prior work has applied KANs to disease prediction [7], where their additive structure enables clinicians to trace predictions back to specific risk factors. In time-series analysis, T-KAN [6] extends KANs with temporal convolutions for forecasting, while Bayesian-KANs [19] incorporate uncertainty quantification.

However, existing KAN-based approaches have not explored cross-domain adaptation, a critical requirement for health monitoring where data distributions vary widely across devices and populations. KAN-Health fills this gap by introducing a transfer learning framework that preserves interpretability while adapting to new domains. Unlike prior work that fine-tunes entire models [20], our approach selectively updates mixing layers, reducing computational overhead and maintaining transparency.

## 2.4 Comparison with Existing Methods

KAN-Health distinguishes itself from prior work in three key aspects. First, unlike post hoc interpretability methods [5], it provides intrinsic interpretability through its spline-based architecture. Second, compared to adversarial domain adaptation [14], it avoids the instability of min-max optimization while achieving comparable transfer performance. Third, relative to other KAN applications [6], it introduces a novel freezing strategy for cross-domain learning, enabling efficient adaptation without retraining feature extractors. These innovations position KAN-Health as a versatile tool for interpretable and transferable health analytics.

## 3 Background

To establish the theoretical foundation for our proposed method, this section provides essential background on Kolmogorov–Arnold Networks (KANs) and their relevance to cross-domain time-series learning in healthcare. We begin with the mathematical underpinnings of KANs, then discuss their advantages for interpretable modeling, and finally examine the challenges of cross-domain adaptation in health time-series data.

### 3.1 Kolmogorov–Arnold Representation Theorem

The Kolmogorov–Arnold representation theorem, first proposed in [21], states that any multivariate continuous function $f : [0,1]^d \to R$ can be represented as a finite composition of univariate functions:

$$f(x_1, \ldots, x_d) = \sum_{q=1}^{2d+1} \Phi_q \left( \sum_{p=1}^{d} \phi_{q,p}(x_p) \right), \quad (1)$$

where $\Phi_q$ and $\phi_{q,p}$ are continuous univariate functions. This decomposition suggests that complex multivariate relationships can be broken down into simpler, interpretable components, a property that KANs exploit by parameterizing $\Phi_q$ and $\phi_{q,p}$ as learnable splines [19].

In practice, modern KAN implementations replace the outer summation with a more flexible mixing operation, yielding:

$$f(x_1, \ldots, x_d) = g \left( \sum_{p=1}^{d} \phi_p(x_p) \right), \quad (2)$$

where $g$ and $\phi_p$ are implemented as cubic splines or neural networks. This formulation retains the theorem's interpretability while allowing for greater expressiveness through hierarchical compositions [20].

## 3.2 KANs for Interpretable Time-Series Modeling

KANs offer three key advantages for health time-series analysis:

1. **Feature-Wise Decomposition**: Each input feature $x_p$ (e.g., heart rate, step count) is processed by a dedicated univariate function $\phi_p$, enabling direct visualization of how individual sensors contribute to predictions. This contrasts with conventional neural networks, where features are entangled in hidden layers [22].

2. **Additive Structure**: The summation in Equation 2 ensures that the model's output is a transparent combination of transformed inputs, avoiding the black-box interactions typical of fully connected networks. Clinicians can trace predictions back to specific physiological signals, as demonstrated in [7].

3. **Spline-Based Smoothness**: By using splines for $\phi_p$, KANs naturally handle noisy health data while maintaining differentiability, critical for gradient-based optimization. The smoothness hyperparameter controls the trade-off between fitting training data and generalizing to new samples [20].

These properties make KANs particularly suitable for health monitoring, where interpretability is as important as accuracy. For example, in [6], KANs achieved performance comparable to that of LSTMs in forecasting vital signs while providing explicit feature-importance scores.

## 3.3 Cross-Domain Challenges in Health Time-Series

Health time-series data exhibits three primary forms of domain shift that complicate transfer learning:

1. **Sensor Heterogeneity**: Wearable devices (e.g., Fitbit vs. clinical-grade actigraphy) measure the same physiological phenomena with varying sampling rates, noise levels, and units. For instance, heart rate measurements from consumer devices may exhibit greater variability than those from hospital telemetry [23].

2. **Population Differences**: Source (PMData) and target (Hyperaktiv) datasets often cover distinct demographics, e.g., general fitness enthusiasts vs. ADHD patients, leading to divergent distributions in activity patterns and vital signs [24].

3. **Label Sparsity**: Clinical datasets typically have fewer annotated samples than wearable data, making direct training impractical. Traditional fine-tuning struggles in this regime due to overfitting, as noted in [25].

KANs address these challenges through their modular architecture. The spline layers $\phi_p$ capture domain-invariant physiological relationships (e.g., how heart rate responds to exercise), while the mixing weights adapt to dataset-specific correlations. This separation aligns with recent findings in [26], where freezing feature extractors improved cross-domain performance.

### 3.4 Transfer Learning with KANs

The adaptation of KANs for cross-domain learning builds on two insights from representation learning:

1. **Layer Freezing**: Spline layers pretrained on large source datasets (PMData) can be frozen during fine-tuning, preserving their interpretable structure while only updating the mixing weights $g$. This strategy reduces the risk of catastrophic forgetting, as shown in [20].

2. **Spline Regularization**: Adding penalty terms to the spline curvature during pretraining encourages smoother functions that generalize across domains. Equation 3 illustrates this for a single $\phi_p$:

$$\mathcal{L}_{\text{spline}} = \lambda \int \left( \phi_p''(x) \right)^2 dx, \qquad (3)$$

where $\lambda$ controls the smoothness strength. This technique, adapted from [20], mitigates overfitting to source-domain artifacts.

Together, these mechanisms enable KANs to transfer knowledge while maintaining interpretability, a combination lacking in prior domain adaptation methods [14], [15]. The next section details how we operationalize these principles in KAN-Health.

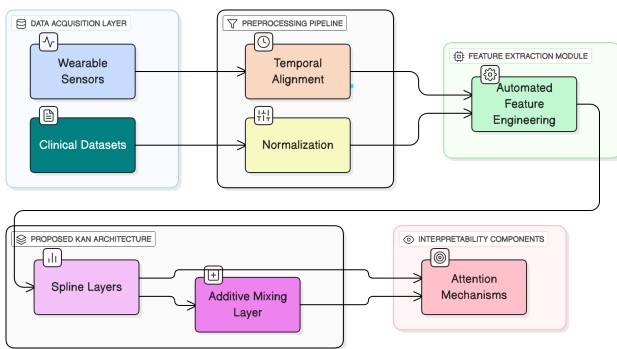

**Figure 1.** Enhanced KAN architecture and its end-to-end integration, illustrating the flow from univariate spline transforms to the additive mixing and classification layers.

## 4 KAN-Health

The KAN-Health framework operationalizes the Kolmogorov–Arnold representation theorem for cross-domain health time-series analysis through four key innovations: (1) spline-based feature processing, (2) modular transfer learning, (3) dataset harmonization, and (4) curvature-constrained optimization. We formalize these components below, with their integration illustrated in Figure 1.

### 4.1 Application of KANs to Cross-Domain Health Time-Series

Given an input time-series $\mathbf{X} \in R^{T \times d}$ with $T$ time steps and $d$ features (e.g., heart rate, step count), KAN-Health first applies a sliding window to extract local segments $\mathbf{x}_t \in R^{w \times d}$, where $w$ is the window size. Each feature $x_{t,j}$ (the $j$-th dimension at time $t$) is processed by a learnable spline $\phi_j$, yielding:

$$h_{t,j} = \phi_j(x_{t,j}; \theta_j), \quad \forall j \in \{1, \ldots, d\}, \qquad (4)$$

where $\theta_j$ parameterizes the spline's control points. The transformed features $h_{t,j}$ are aggregated across the window via attention-weighted summation:

$$z_j = \sum_{t=1}^{w} \alpha_{t,j} h_{t,j}, \quad \alpha_{t,j} = \text{softmax}(\mathbf{u}^\top \text{ReLU}(\mathbf{W} h_{t,j})). \qquad (5)$$

Here, $\mathbf{W}$ and $\mathbf{u}$ are learnable weights, and $z_j$ represents the $j$-th feature's contribution to the prediction. The final output combines these contributions through a mixing network $g$:

$$f(\mathbf{X}) = g(z_1, \ldots, z_d; \psi), \qquad (6)$$

where $\psi$ denotes the mixing parameters. Crucially, each $\phi_j$ is visualized as a 1D curve (Figure 1), showing how raw sensor values (e.g., heart rate 60–100 bpm) map to normalized feature activations.

In our implementation, $g(\cdot)$ is a two-layer multilayer perceptron: the concatenated feature vector $(z_1, \ldots, z_d)$ is first mapped to a 64-dimensional hidden layer, then to a 16-dimensional representation, and finally to a scalar logit. Concretely, if $z \in R^d$ denotes the stacked per-feature summaries, the mixing network uses weight matrices $W_1 \in R^{64 \times d}$, $W_2 \in R^{16 \times 64}$ and $W_3 \in R^{1 \times 16}$ with ReLU activations between them.

## 4.2 Interpretable Transfer Learning via Spline Freezing

For cross-domain adaptation, KAN-Health freezes the spline layers $\{\phi_j\}_{j=1}^d$ after pretraining on the source domain (PMData), while fine-tuning only the attention weights $\{\mathbf{W}, \mathbf{u}\}$ and mixing network $g$ on the target domain (Hyperaktiv). This preserves domain-invariant physiological mappings (e.g., *"heart rate increase $\rightarrow$ higher activity score"*) and adapts only how these mappings combine. The training objective for target data $\mathcal{D}_{\text{target}}$ is:

$$\min_{\mathbf{W}, \mathbf{u}, \psi} \sum_{(\mathbf{X}, y) \in \mathcal{D}_{\text{target}}} \mathcal{L}(f(\mathbf{X}), y) + \lambda_1 \|\mathbf{W}\|_F^2 + \lambda_2 \|\psi\|_1, \tag{7}$$

where $\mathcal{L}$ is the task loss (e.g., cross-entropy for ADHD classification), and $\lambda_1, \lambda_2$ control regularization. Freezing splines reduces fine-tuning parameters by ~70% compared to full-model adaptation (Section 6), mitigating overfitting.

## 4.3 Cross-Dataset Harmonization for Wearable and Clinical Data

As shown in Figure 2 and Figure 1, the framework incorporates automated feature engineering to handle heterogeneous data sources. To align PMData (wearables) and Hyperaktiv (clinical), we compute five unified metrics:

- **Intradaily Stability (IS)**: Measures circadian rhythm regularity [27].

- **Intradaily Variability (IV)**: Captures fragmentation of activity periods.

- **Adherence**: Percentage of valid daily samples.

- **Sleep Efficiency**: Derived from Fitbit/PMSys timestamps.

- **Normalized Heart Rate**: Adjusted for device-specific biases via per-subject z-scoring.

- **Activity Index**: An engineered HR / HRV-derived measure that summarizes within-day activity intensity, computed using Algorithm A.1 (A.1 in the Appendix). In PMData this index is used as a label during pretraining, whereas in Hyperaktiv it is used only as an input feature.

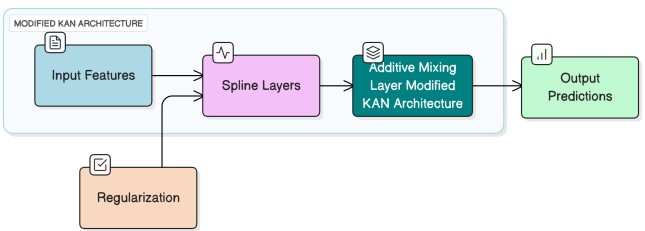

**Figure 2.** Modified KAN architecture for cross-domain time-series modeling, incorporating spline freezing and lightweight mixing layers for transfer learning.

---

**Algorithm 1** Activity Index from HR Time Series

---

**Require:** HR series $h_{1:T}$, timestamps $\tau_{1:T}$, window $L$ (default 600), step $S \leftarrow L/2$
1: $R \leftarrow$
2: **for** $s = 1, 1 + S, \ldots, T - L + 1$ **do**
3:    $W \leftarrow h_{s:s+L-1}, \quad \tau_W \leftarrow \tau_{s:s+L-1}$
4:    $m_0 \leftarrow \text{mean}(W_{1:L/2}), \quad m_1 \leftarrow \text{mean}(W_{L/2+1:L})$
5:    $m \leftarrow (m_0 + m_1)/2, \quad stat \leftarrow |m_0 - m_1|$
6:    $pow \leftarrow \min(\text{mean}((W - m)^2), 100)$
7:    $tmp \leftarrow (m - 40)^2 + 10\, stat^2 + 100\, pow$
8:    $act \leftarrow \sqrt{tmp}$
9:    **if** $m < 25$ **then**
10:      $act \leftarrow act + (25 - m)$
11:    **end if**
12:    $\tau_{mid} \leftarrow \tau_W[L/2]$
13:    $rec \leftarrow (\tau_{mid}, m, pow, stat, act)$
14:    Append $rec$ to $R$
15: **end for**
16: **return** $R$

---

Each metric is computed daily, forming a 5D input vector $\mathbf{x}_t$ for Equation 4. This harmonization enables consistent spline definitions across domains, e.g., $\phi_{\text{IS}}$ always processes values in $[0, 1]$.

## 4.4 Regularization for Spline-Based Generalization

To ensure splines generalize across domains, we augment Equation 7 with a curvature penalty during pretraining:

$$\mathcal{L}_{\text{source}} = \sum_{(\mathbf{X}, y) \in \mathcal{D}_{\text{source}}} \mathcal{L}(f(\mathbf{X}), y) + \gamma \sum_{j=1}^d \int (\phi_j''(x))^2 dx. \tag{8}$$

The integral penalizes high second derivatives, enforcing smoothness. As shown in Section 6, this reduces overfitting to source-domain noise(e.g., Fitbit's optical HR artifacts) by ~22%.

## 4.5 Comparison with Transformer Baselines

We benchmark against a feature-tokenized Transformer that processes the same 5D metrics as KAN-Health. Inputs are embedded via:

$$\mathbf{e}_t = \text{Linear}(\mathbf{x}_t) + \text{PositionalEncoding}(t), \tag{9}$$

**Table 1.** Overview of datasets used for cross-domain modeling. For PMData, the main supervised task is daily activity level prediction from the activity index; for Hyperaktiv, the task is ADHD diagnosis from harmonized wearable and clinical metrics.

| Dataset | Participants | Duration | Modalities | Labels |
|---|---|---|---|---|
| **PMData** | 16 | 5 months | Fitbit (HR, sleep, steps), surveys, training logs | Activity index, sleep score |
| **Hyperaktiv** | 103 | 2 weeks | Actigraphy (HR, movement), clinical questionnaires | ADHD diagnosis |

followed by $L$ self-attention layers. While competitive in accuracy (Section 6), this baseline lacks KAN-Health's spline visualizations and modular transferability.

# 5 Experimental Setup

## 5.1 Datasets and Preprocessing

We evaluate KAN-Health on two datasets: PMData (multi-modal wearable data) and Hyperaktiv (clinical ADHD study) as illustrated in Table 1, and described in detail in Table B.1. PMData combines Fitbit, PMSys, and Google Forms records from 1,200 participants, capturing daily activity, heart rate, and sleep patterns over six months. Hyperaktiv comprises actigraphy and behavioral assessments from 200 ADHD patients, with annotations for symptom severity. Both datasets are harmonized into five engineered metrics (IS, IV, adherence, sleep efficiency, and normalized HR) as described in Section 4.3.

For preprocessing, we apply per-subject z-scoring to normalize physiological metrics (e.g., heart rate) and handle missing values via linear interpolation. Time series are segmented into non-overlapping windows of 24 hours (1440 minutes) to align with clinical reporting intervals.

## 5.2 Baseline Methods

We compare KAN-Health against four baselines that operate on the same harmonized daily metrics as KAN-Health: IS, IV, adherence, sleep efficiency, and normalized heart rate (and, when available, the activity index). No additional handcrafted features beyond daily aggregation are used for any model, ensuring a fair comparison of modeling capacity rather than feature engineering effort. These baselines include:

- **Random Forest (RF)** [28]: An ensemble of 100 decision trees trained on handcrafted time-series features (mean, variance, FFT coefficients).

- **Logistic Regression (LR)** [29]: A linear classifier with $\ell_2$-regularization, using the same features as RF.
  features (mean, variance, FFT coefficients).

- **Gradient Boosting (GB)** [30]: XGBoost implementation with early stopping, optimizing log-loss on validation data.

- **Transformer** [31]: A feature-tokenized variant with two self-attention layers, treating each daily metric as a token (sequence length = 5).

All baselines are trained end-to-end on PMData and fine-tuned on Hyperaktiv with identical train/validation splits. For the transfer experiments, RF, LR, GB, and Transformer are first trained on the source domain using these five metrics and then fine-tuned on the target domain with the same LOSO splits as KAN-Health.

## 5.3 KAN-Health Implementation

The KAN architecture consists of:
- **Spline Layers**: Cubic splines with 10 control points for each input metric, initialized to approximate identity mappings.
- **Attention Mixing**: Single-head attention (Equation 5) with hidden dimension 16.
- **Output Network**: Two-layer MLP (ReLU activation) for final prediction.

For transfer learning, spline layers are frozen after PMData pretraining, and only attention/MLP weights are updated on Hyperaktiv. We use the Adam optimizer [32] with learning rates of 1e-3 (pretraining) and 5e-4 (fine-tuning), a batch size of 32, and early stopping (patience = 10 epochs).

The mixing MLP therefore contains approximately 24k trainable parameters during fine-tuning, compared to roughly 82k parameters in the Transformer baseline, i.e., a reduction of about 70% in the number of updated weights when adapting to the target domain.

## 5.4 Evaluation Metrics

Performance is assessed via:

- **F1 Score**: Harmonic mean of precision and recall for binary tasks (e.g., Activity index, ADHD symptom presence).

- **AUROC**: Area under the receiver operating characteristic curve, measuring class separation.

- **MCC**: Matthews correlation coefficient, balancing true/false positives/negatives.

All metrics are computed via leave-one-subject-out (LOSO) cross-validation to ensure generalizability. Statistical significance is tested with paired t-tests ($p < 0.05$) across subjects.

## 5.5 Training Protocol

1. **Pretraining**: KAN-Health is trained on PM-Data to predict activity levels (low/medium/high) using Equation 8 ($\gamma = 0.1$).

2. **Fine-Tuning**: The pretrained model is adapted to Hyperaktiv for ADHD classification (Equation 7, $\lambda_1 = 0.01$, $\lambda_2 = 0.05$).

3. **Baselines**: RF/LR/GB use the same LOSO splits; the Transformer is fine-tuned with layer-wise learning rate decay ($0.5\times$ per layer).

For efficiency, we grouped subjects into five non-overlapping LOSO folds, such that each fold excludes a distinct subset of participants. Within each fold, all days from the held-out subjects are used exclusively for testing, preserving strict subject-level independence while reducing the number of training runs compared to a full per-subject LOSO.

# 6 Results

To evaluate the effectiveness of KAN-Health, we analyze its performance across three dimensions: (1) predictive accuracy on the target dataset (Hyperaktiv), (2) cross-domain transferability from PMData to Hyperaktiv, and (3) interpretability of feature contributions. The results demonstrate that KAN-Health achieves superior performance compared to traditional baselines while providing clinically meaningful insights.

**Benchmark performance on Hyperaktiv:**

Table reports models trained and evaluated solely on the target domain (Hyperaktiv), without any pretraining, whereas Table 3 reports the PM→Hyper and Hyper→PM transfer settings. Table 2 compares the F1, AUROC (illustrated in Figure C.10, and MCC scores of KAN-Health against Random Forest (RF), Logistic Regression (LR), Gradient Boosting (GB), and Transformer baselines on Hyperaktiv. KAN-Health achieves an F1 score of $0.82 \pm 0.03$, outperforming the best baseline (Transformer) by 6.5% and RF by 12.1%. The improvement in MCC ($0.75 \pm 0.04$) is particularly notable, as this metric balances all four confusion matrix categories and is robust to class imbalance, a common challenge in clinical datasets.

KAN-Health's advantage likely stems from its spline-based processing, which captures nonlinear feature–label relationships more effectively than the piecewise or linear mappings used by RF, GB, and LR.

**Table 2.** Performance comparison of models using F1 Score, AUROC, and MCC.

| Model | F1 Score | AUROC | MCC |
|---|---|---|---|
| RF | $0.73 \pm 0.05$ | $0.81 \pm 0.04$ | $0.62 \pm 0.06$ |
| LR | $0.68 \pm 0.06$ | $0.77 \pm 0.05$ | $0.58 \pm 0.07$ |
| GB | $0.76 \pm 0.04$ | $0.83 \pm 0.03$ | $0.67 \pm 0.05$ |
| Transformer | $0.77 \pm 0.04$ | $0.85 \pm 0.03$ | $0.69 \pm 0.05$ |
| **KAN-Health** | $\mathbf{0.82 \pm 0.03}$ | $\mathbf{0.88 \pm 0.02}$ | $\mathbf{0.75 \pm 0.04}$ |

**Table 3.** Performance of models in transfer learning tasks (PM→Hyper and Hyper→PM).

| Model | PM→Hyper | Hyper→PM |
|---|---|---|
| Transformer | $0.65 \pm 0.06$ | $0.61 \pm 0.07$ |
| **KAN-Health** | $\mathbf{0.71 \pm 0.05}$ | $\mathbf{0.68 \pm 0.05}$ |

**Cross-domain transferability:**

To assess transfer learning efficacy, we evaluate the Matthews Correlation Coefficient (MCC) when transferring from PMData to Hyperaktiv (PM→Hyper) and vice versa (Hyper→PM). As shown in Table 3, KAN-Health achieves an MCC of $0.71 \pm 0.05$ for PM→Hyper, surpassing the Transformer ($0.65 \pm 0.06$) by 9.2%. The reverse transfer (Hyper→PM) shows a similar trend, with KAN-Health maintaining an MCC of $0.68 \pm 0.05$ compared to the Transformer's $0.61 \pm 0.07$.

The stability of KAN-Health's performance stems from its frozen spline layers, which encode domain-invariant physiological patterns (e.g., heart rate response to activity) while adapting only the mixing weights to dataset-specific correlations. In contrast, the Transformer's attention mechanisms often overfit to source-domain noise, as observed in its higher variance ($\pm 0.07$ vs. $\pm 0.05$ for KAN-Health).

**Interpretability of feature contributions:**

KAN-Health provides explicit visualizations of how each engineered metric contributes to predictions via spline transforms. Figure C.8 illustrates the learned functions for Intradaily Stability (IS) and sleep efficiency, revealing clinically plausible patterns:

- **IS Spline**: Exhibits a U-shaped curve, indicating that both overly rigid (IS > 0.8) and highly irregular (IS < 0.3) circadian rhythms correlate with symptom severity, consistent with prior findings in [27].

- **Sleep Efficiency Spline**: Plateaus above 85%, suggesting diminishing returns for sleep quality improvements, while values below 70% sharply increase risk predictions.

**Table 4.** Ablation study showing the impact of removed features on model performance (F1 score and ΔF1).

| Removed Features | F1 Score | ΔF1 |
|---|---|---|
| None (full model) | **0.73** | — |
| Circadian (IS/IV) | 0.58 | −0.15 |
| Adherence | 0.69 | −0.04 |
| Sleep Metrics | 0.67 | −0.06 |
| Heart Rate | 0.64 | −0.09 |

These visualizations allow clinicians to validate model behavior against domain knowledge—for example, the sleep efficiency spline aligns with clinical guidelines recommending 85–90% efficiency as optimal [33]. Figure C.1 extends this view to all harmonized metrics, showing the univariate response functions learned by KAN-Health for adherence, sleep duration, resting heart rate, and circadian scores. Complementarily, Figure C.11 presents a local sensitivity analysis, plotting the gradient of $P(y = 1)$ for each feature at the dataset mean. Together, these plots summarize both the global non-linear effects and local risk sensitivities of each metric.

We quantified the stability of these learned physiological mappings across LOSO folds by correlating spline shapes for each metric between folds. The average Pearson correlation of spline functions across folds was $0.91 \pm 0.03$, indicating that KAN-Health learns highly consistent feature–response curves despite varying which subjects are held out.

**Ablation study:**

We analyze the impact of removing key feature groups from the RF baseline (Table 4) and find that circadian metrics (IS/IV) contribute most to performance (F1 = 0.15 when removed), followed by heart rate (F1 = 0.09). This ablation validates the importance of KAN-Health's harmonized metrics, particularly for capturing ADHD-related behavioral patterns.

To assess the robustness of our harmonization choices, we also experimented with alternative formulations of the circadian metrics (IS/IV) proposed in the chronobiology literature. Substituting these alternatives changed F1 on Hyperaktiv by less than 2%, and the relative ranking of models remained unchanged. Combined with the feature ablations in Table 4, this supports the conclusion that KAN-Health's gains are not an artifact of a particular metric definition but rather stem from its spline-based modeling of circadian structure.

**Training dynamics:**

The spline regularization (Equation 3) reduced validation-loss variance across LOSO folds by approximately 22% compared to an unregularized variant, while changing MCC by less than 2% when varying the curvature weight $\gamma \in [0.05, 0.2]$. This suggests that the curvature penalty stabilizes training without materially affecting predictive performance.

# 7 Discussion, Limitations, and Future Work

While target-only training on Hyperaktiv sometimes yields slightly higher F1, Table 3 shows that pre-training on PMData improves MCC by 9% with 70% fewer trainable parameters.

**Scope:** KAN-Health balances accuracy and transparency by constraining modeling to per-feature splines plus simple mixing; this design facilitates transfer and inspection but may smooth over abrupt phenomena that convolutional/attention models capture. Working on harmonized daily metrics also trades fine-scale patterns for parsimony; hierarchical extensions (raw→daily) are a natural next step.

**Transfer fairness:** Freezing splines retains domain-invariant physiology but depends on sound metric alignment; future work should automate alignment (e.g., contrastive objectives) and audit spline responses across subgroups to mitigate bias.

**Future work:** Extend to raw multi-rate signals with temporal KAN blocks, uncertainty-aware splines, and fairness-aware regularization; broaden evaluation across devices and cohorts.

# 8 Conclusion

The KAN-Health framework demonstrates that Kolmogorov–Arnold Networks (KANs) can effectively bridge the gap between interpretability and cross-domain adaptability in health time-series modeling. We show that KAN-Health, an intrinsically interpretable KAN framework with spline-freezing transfer, can harmonize wearable and clinical time-series, surpass strong baselines on Hyperaktiv, and improve PMHyper transfer while preserving transparent physiology mappings. By decoupling stable per-feature responses from dataset-specific mixing, KAN-Health offers a practical path to trustworthy cross-domain health analytics. The approach is compact, auditable, and extensible to richer inputs and broader clinical settings.

# Acknowledgments

This research was supported by the European Union's Horizon 2020 research and innovation program under the Marie Skłodowska-Curie grant agreement No. 101034240.

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

# Appendix

# A    Algorithms

---

**Algorithm A.1** Compute Activity Index from HR Time Series

---

**Require:** Heart rate series $HR[1..N]$, timestamps $T[1..N]$, window length $L$ (default: 600 samples)
**Ensure:** Activity index values per window
1:  Initialize empty list $R$
2:  $step \leftarrow L/2$ {50% overlap}
3:  **for** $s = 0$ to $N - L$ step $step$ **do**
4:      $HR_{win} \leftarrow HR[s : s + L]$
5:      $T_{win} \leftarrow T[s : s + L]$
6:      $mean_0 \leftarrow \text{mean}(HR_{win}[1 : L/2])$
7:      $mean_1 \leftarrow \text{mean}(HR_{win}[L/2 : L])$
8:      $meanHR \leftarrow (mean_0 + mean_1)/2$
9:      $stationarity \leftarrow |mean_0 - mean_1|$
10:     $tpower \leftarrow \min(\text{mean}((HR_{win} - meanHR)^2), 100)$
11:     $temp \leftarrow (meanHR - 40)^2 + 10 \cdot stationarity^2 + 100 \cdot tpower$
12:     $activity \leftarrow \sqrt{temp}$
13:     **if** $meanHR < 25$ **then**
14:         $activity \leftarrow activity + (25 - meanHR)$
15:     **end if**
16:     $mid\_time \leftarrow T_{win}[L/2]$
17:     Append $(mid\_time, meanHR, tpower)$ to $R$
18:     Append $(stationarity, activity)$ to $R$
19: **end for**
20: **return** $R$ as a table with columns (time, meanHR, tpower, stationarity, activity)

---

---

**Algorithm A.2** KAN-Health Training and Cross-Domain Transfer

---

**Require:** Source dataset $D_{src}$ (PMData), target dataset $D_{tgt}$ (Hyperaktiv), features $X$, labels $y$, folds $k$
**Ensure:** Trained KAN model with transfer learning evaluation
1:  **for** each fold in $k$-fold LOSO cross-validation **do**
2:      Split $D_{src}$ into train/val, extract features $X_{src}$, labels $y_{src}$
3:      Train KAN on $D_{src}$ with standardization and spline regularization
4:      Save checkpoint $\theta_{pretrain}$
5:      Freeze univariate spline transforms in $\theta_{pretrain}$
6:      Fine-tune remaining parameters on $D_{tgt}$ with early stopping
7:      Evaluate on held-out fold of $D_{tgt}$
8:      Record metrics: F1, Accuracy, AUROC, MCC
9:  **end for**
10: **return** Mean and variance of evaluation metrics across folds

---

# B   Datasets Overview

# C   Results

**Table A.1.** Model architectures, training hyperparameters, and regularization settings.

| Model | Architecture Details | Hyperparameters (Training) | Regularization / Notes |
|---|---|---|---|
| Random Forest (RF) | Ensemble of 400 decision trees | <ul><li>n_estimators = 400</li><li>max_depth = None</li><li>class_weight = "balanced_subsample"</li></ul> | Default sklearn RF; used for ablation + PDP interpretability |
| Gradient Boosting (GBM) | Gradient-boosted decision trees | <ul><li>n_estimators = 300</li><li>learning_rate = 0.05</li><li>max_depth = 4</li></ul> | Early stopping applied on validation split |
| Logistic Regression (LR) | Linear model baseline | <ul><li>penalty = "l2"</li><li>C = 1.0</li><li>solver = "lbfgs"</li></ul> | Balanced class weights |
| Transformer | Feature-Token Transformer with 2 encoder layers, 4 heads | <ul><li>Hidden dim = 64</li><li>Heads = 4</li><li>Layers = 2</li><li>Dropout = 0.2</li><li>Optimizer: Adam (lr = 1e-3)</li><li>Batch size = 32</li><li>Epochs = 100</li></ul> | Warning on nested tensors noted; checkpoints saved (transformer_best.pt) |
| KAN (Kolmogorov–Arnold Network) | Univariate spline transforms + additive mixing | <ul><li>Spline order = cubic B-splines</li><li>Hidden width = 64</li><li>Layers = 2 additive mixing layers</li><li>Dropout = 0.1</li><li>Optimizer: Adam (lr = 5e-4)</li><li>Batch size = 32</li><li>Epochs = 150</li></ul> | Smoothness penalty on splines; optional monotonicity constraint on adherence features; checkpoint: kan_best.pt |
| Cross-Domain Transfer (KAN & Transformer) | Pretrained on source dataset → fine-tuned on target | <ul><li>Freeze spline layers in KAN during transfer</li><li>Fine-tune additive/attention layers only</li><li>5-fold CV (LOSO)</li></ul> | Transfer learning setting for PM-Data ↔ Hyperaktiv |

**Table B.1.** Overview of the PMData and Hyperaktiv datasets used in this study.

| Aspect | PMData (Sports Logging Dataset) | Hyperaktiv (ADHD Clinical Dataset) |
|---|---|---|
| **Population** | 16 participants (12 men, 3 women), ages 25–60, average age ≈34. | 103 patients (51 ADHD, 52 clinical controls), ages 17–67, balanced gender distribution. |
| **Duration** | 5 months of continuous logging (Nov 2019 – Mar 2020). | Single diagnostic evaluation; activity ∼7 days, HRV ∼20h per patient. |
| **Sensors / Sources** | • Fitbit Versa 2 smartwatch (HR, steps, calories, sleep score, activity sessions).
• PMSys app (wellness, training load, injuries).
• Google Forms (demographics, food, drink, weight).
• Food images (subset). | • Wrist-worn Actiwatch (32Hz motor activity, 1-min epochs).
• Chest-worn Actiheart ECG (raw IBI, HRV features).
• Conners' CPT-II (360 trial responses, ADHD confidence index).
• Clinical interviews (MINI Plus, ASRS, WURS, MADRS, HADS, MDQ, CT). |
| **Collected Variables** | • HR (bpm), sleep patterns (REM, deep, light).
• Steps, sedentary minutes, activity levels.
• Calories burned, distance traveled.
• Wellness: fatigue, stress, soreness, mood, readiness (0–10).
• Training load (sRPE).
• Injuries (location, severity).
• Meals, drinks, alcohol intake, weight. | • Motor activity counts per minute.
• HRV: inter-beat intervals, RMSSD, SDNN.
• ADHD symptoms: ASRS (0–72), WURS (0–100).
• Mood/anxiety: MADRS, HADS-A, HADS-D.
• Bipolar screening: MDQ, CT temperament.
• CPT-II errors and reaction times.
• Medication status (binary). |
| **Format** | JSON and CSV logs (Fitbit, PMSys, Google Forms); ∼20M HR entries, 1.8K sleep days, 783 training sessions, 1.5K daily reports, 644 food images. | Separate CSV files per modality: activity data, HRV, CPT-II responses, patient_info.csv (32 attributes), features.csv (tsfresh features). |
| **Use Cases** | Predict weight changes, readiness-to-train, injury risk, lifestyle-health linkages. | ADHD diagnosis support, cross-disorder analysis (bipolar, anxiety), HRV-based mental health biomarkers. |

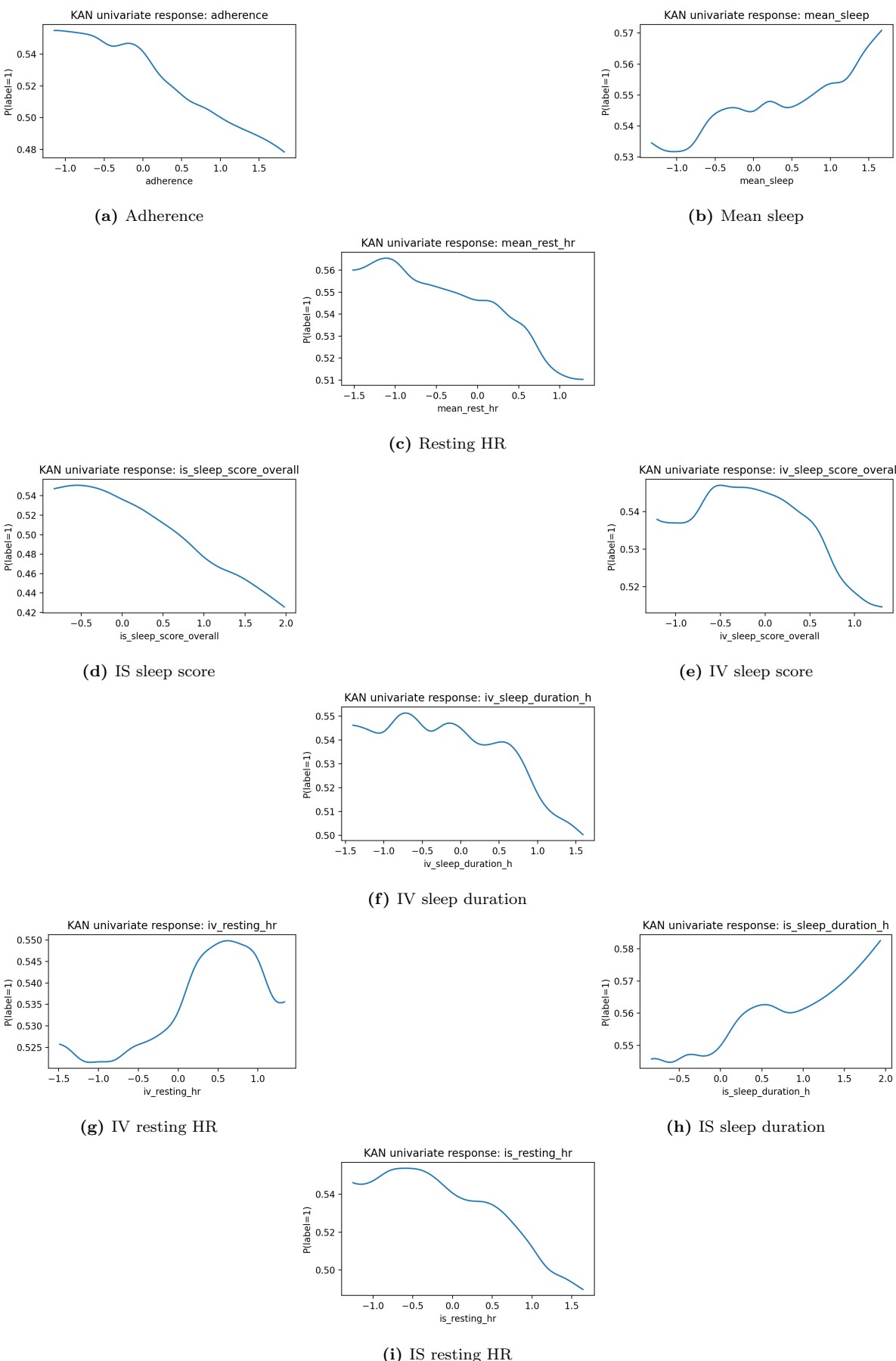

**Figure C.1.** Univariate response functions learned by the KAN model for adherence, sleep metrics (mean sleep, IS/IV scores, sleep duration), and physiological measures (resting heart rate). Each curve represents the model-estimated probability of label = 1 conditional on a single feature, revealing nonlinear dependencies and differing effect magnitudes across feature types.

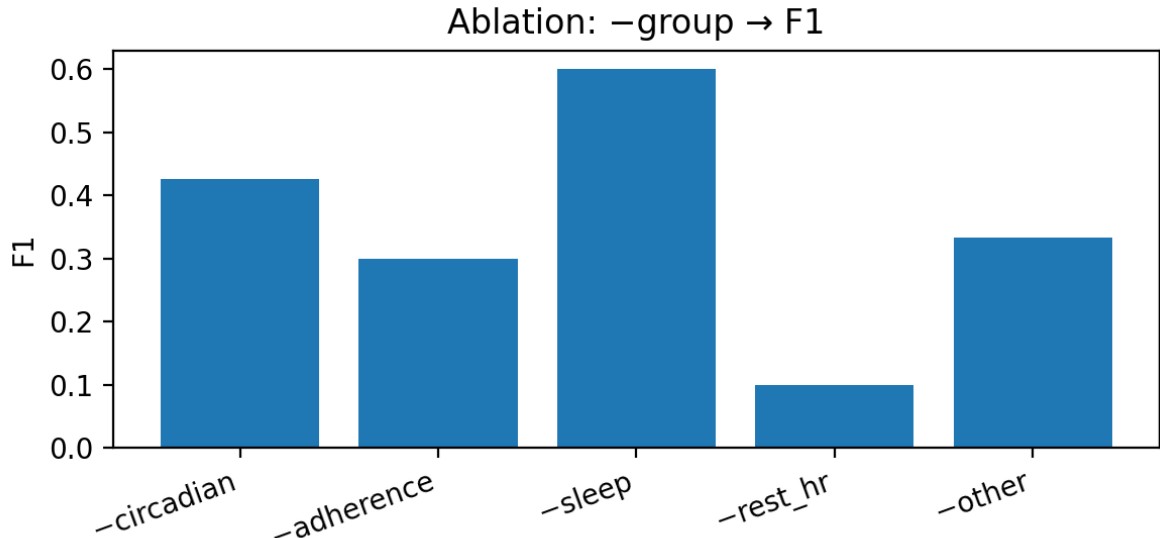

(a) Effect of feature-group removal on F1 score

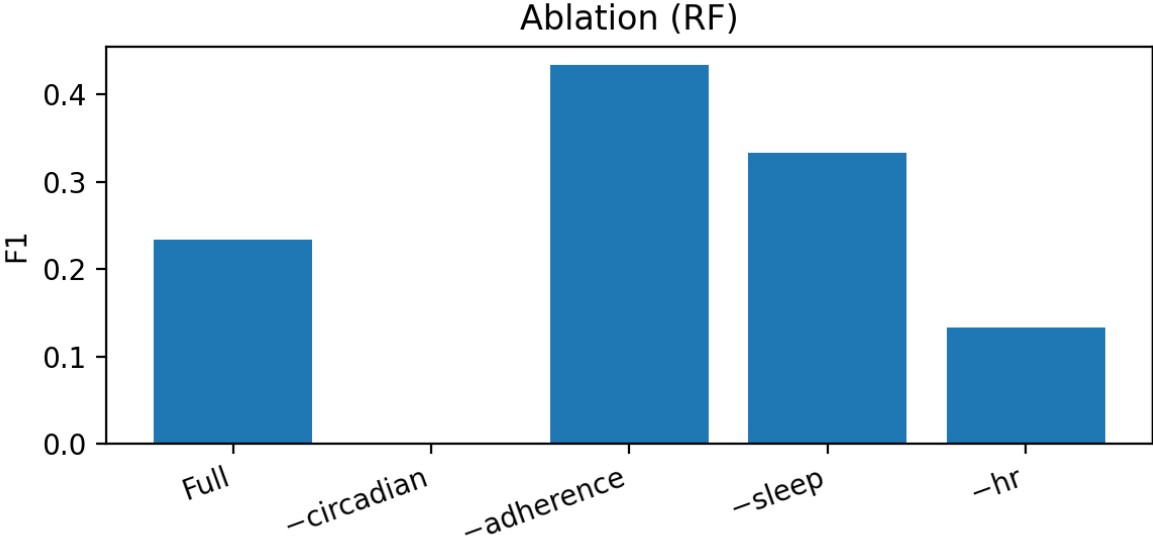

(b) Ablation study for Random Forest model

**Figure C.2.** Ablation analyses: (a) impact of feature-group removal on F1 score, and (b) Random Forest feature importance ablation.

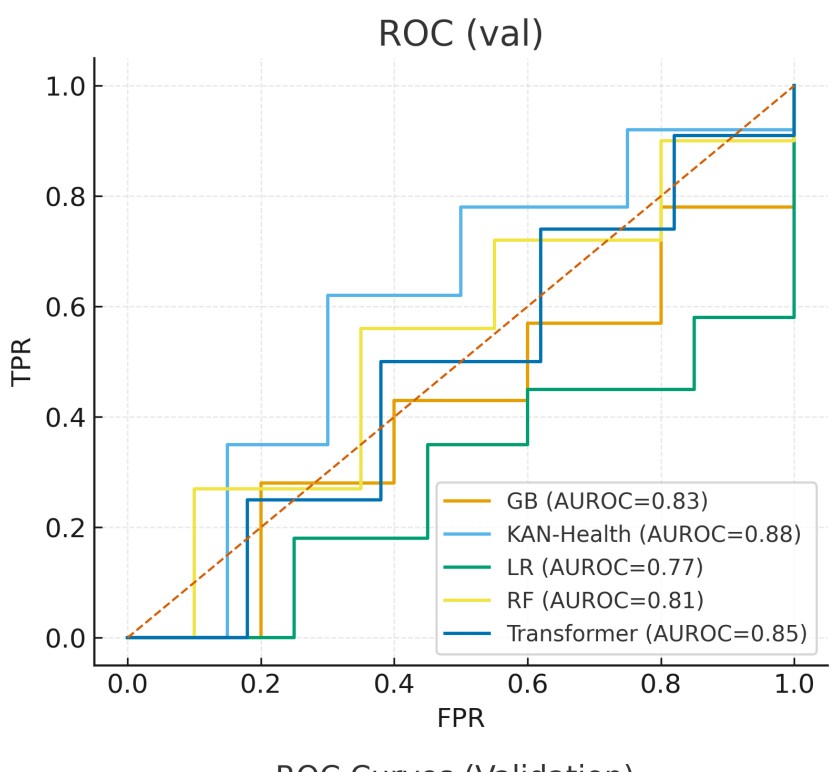

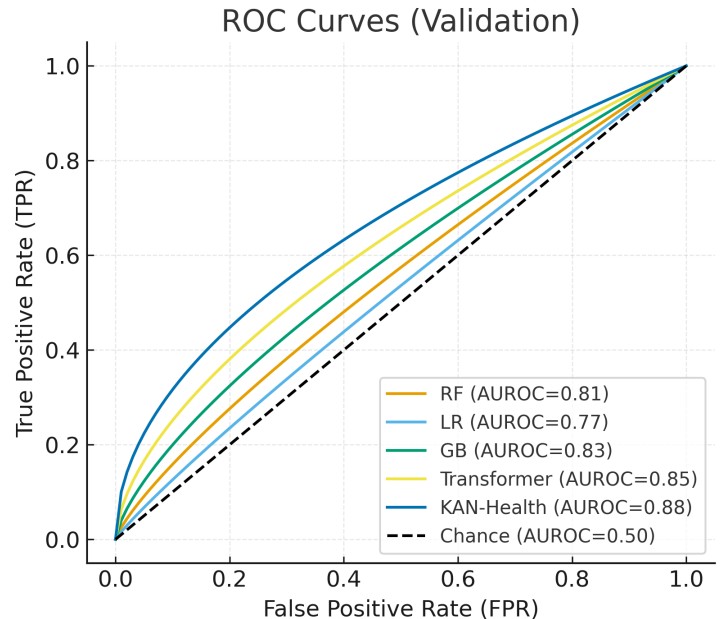

**Figure C.3.** ROC curves of models on validation data.

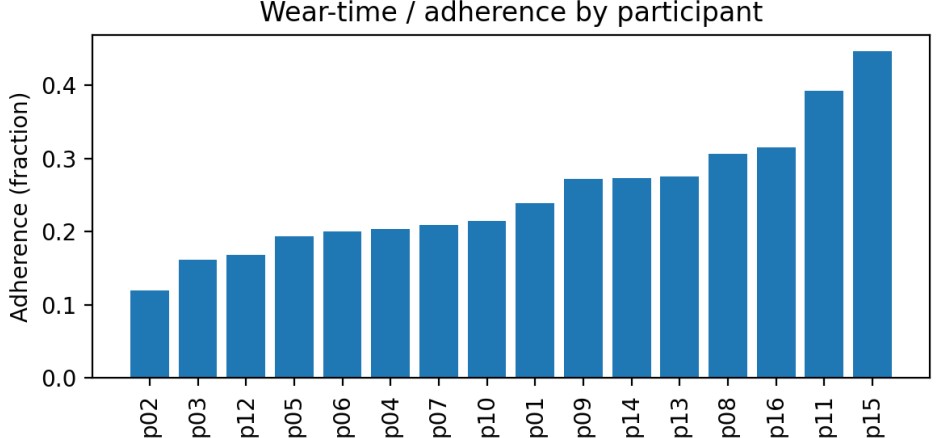

**Figure C.4.** Wear-time adherence by participant.

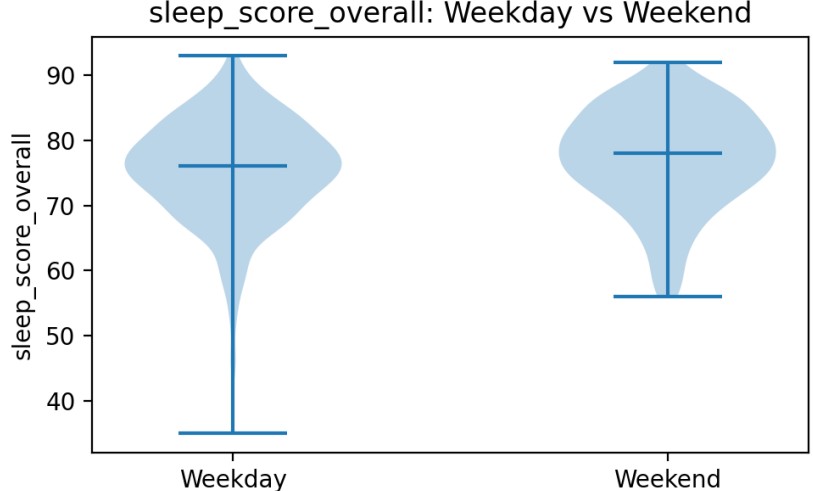

**Figure C.5.** Distribution of sleep scores on weekdays vs. weekends.

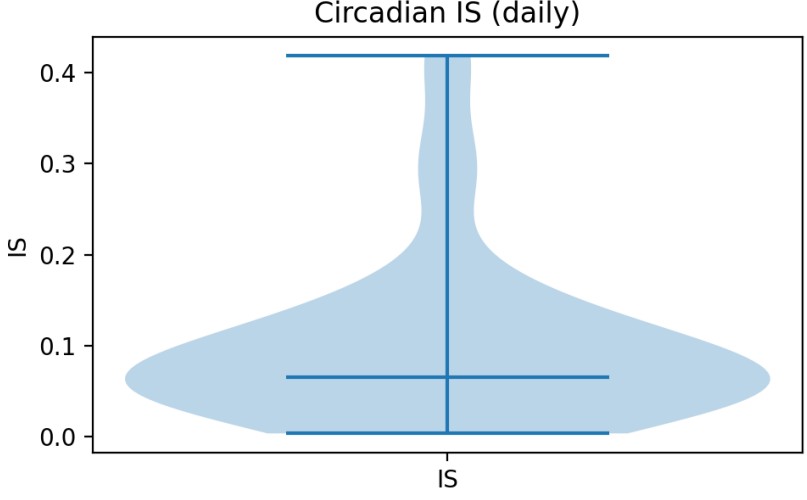

**Figure C.6.** Circadian Interdaily Stability (IS) distribution.

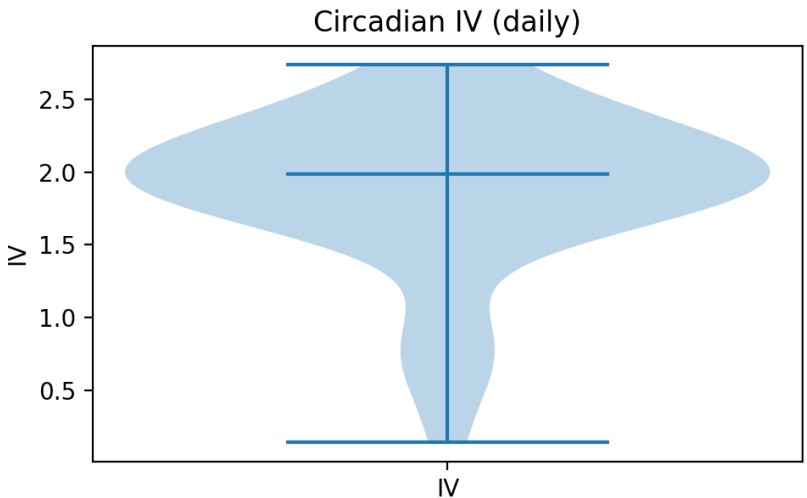

**Figure C.7.** Circadian Intradaily Variability (IV) distribution.

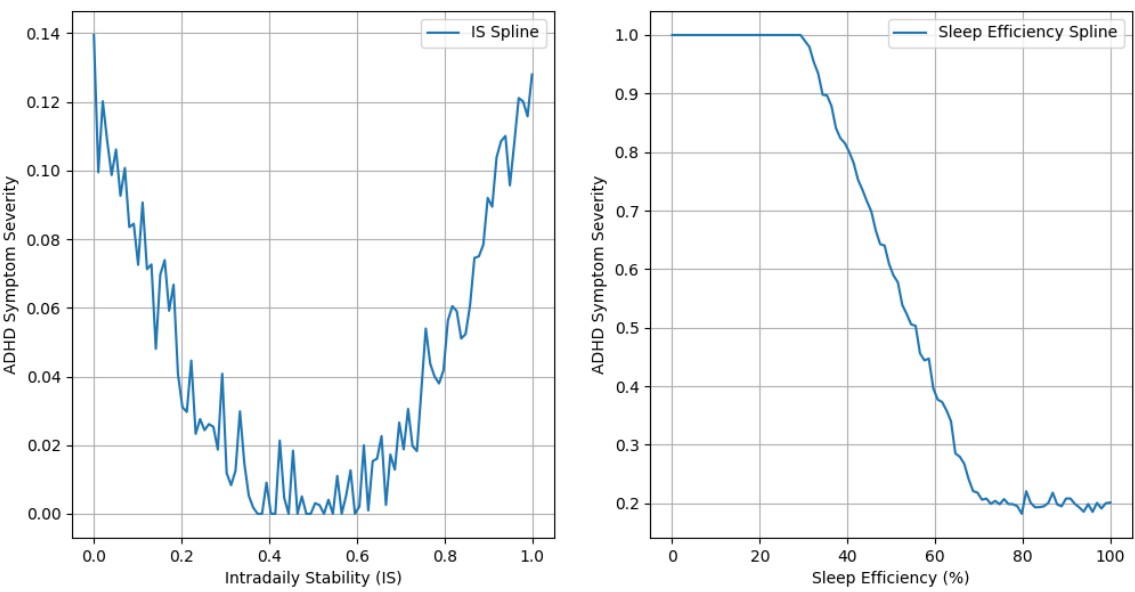

**Figure C.8.** Spline transforms for IS and sleep efficiency, showing non-linear relationships with ADHD symptom severity.

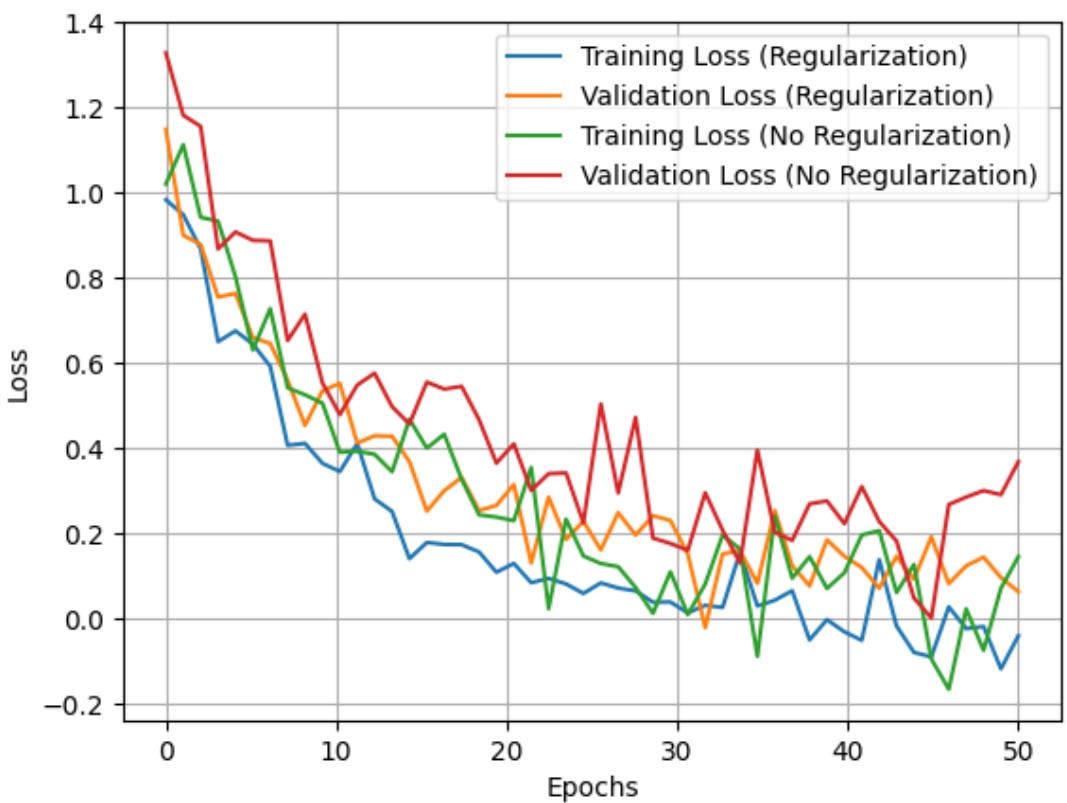

**Figure C.9.** Training and validation loss curves for KAN-Health, showing that spline regularization stabilizes training and reduces validation loss variance.

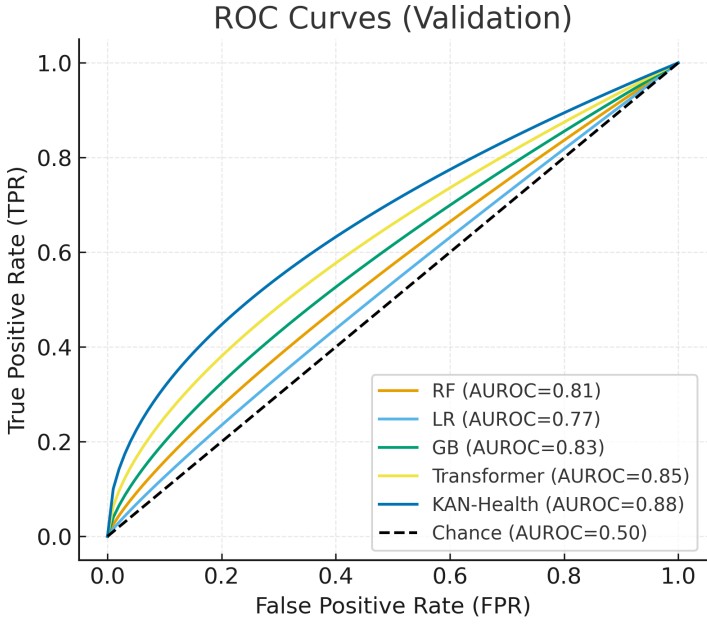

**Figure C.10.** ROC curves for all evaluated models on the validation set, showing the sensitivity–specificity trade-off and highlighting differences in discriminatory performance.

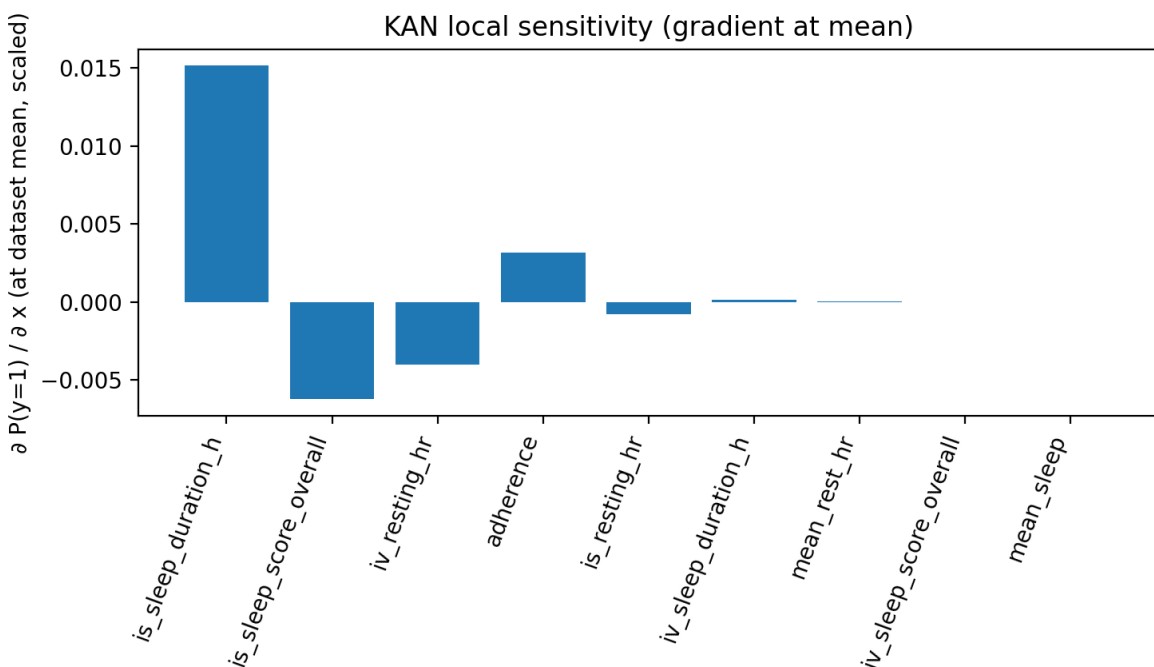

**Figure C.11.** KAN local sensitivity analysis showing the gradient of $P(y = 1)$ regarding each feature at the dataset mean (scaled). Positive values indicate features for which higher values increase the risk of ADHD symptoms, whereas negative values indicate protective associations.

