# OpenReview forum: "Kolmogorov–Arnold Networks for Cross-Domain Time-Series Modeling in Health and Activity Monitoring"
_NLDL.org/2026/Conference — NLDL 2026 Oral_

### Official Review · Reviewer_QeyQ · 2025-09-30
**Very clear illustration of Kolmogorov-Arnold Networks applied to supervised learning from time-series health data**

**Rating:** 5
**Confidence:** 4
**Final Rating:** 5
**Final Confidence:** 4

**Summary:**

This paper presents an application of Kolmogorov-Arnold Networks to supervised learning problems  using time-series health data. KANs differ from standard feedforward networks in that their free weights parametrise (spline-based) activation functions, which are therefore learned. An architecture based on the Kolmogorov-Arnold representation theorem also offers a way for features to be given visual interpretation. There is also an investigation into a transfer learning problem, demonstrating the robustness of this approach to variations in the way data are recorded and to the chosen task.

**Strengths:**

Clarity and presentation: This is the model of a well-written paper. It is mathematically precise, motivations are given clearly, and the contributions of this paper over existing work are well explained. Experiments are sensible; results are well presented and validate the main messages of the work.

Coherence and correctness: I found no errors. The paper gives a clear demonstration of the potential for KANs in the health data domain, where interpretability is at a premium and cross-domain modelling via transfer learning is desirable. There is a set of empirical experiments with comparison against a reasonable collection of other supervised approaches, including a simple transformer architecture. Certain features are given biological interpretation (Fig 3) which I especially liked.

Novelty: KANs have recently become very popular and there appears to have been little previous work in the way of application of KANs in this context. Specific challenges when it comes to working with multi-modal health data are addressed, and there is some discussion of how two different datasets can be reconciled.

Reproducibility: There is a placeholder for a URL which will host code used for the experiments. I expect this to be completed on acceptance, so there should be no issues with reproducibility.

Discussion of limitations: This is included and contains a list of obvious desirable extensions.

**Weaknesses:**

These are only minor comments and should be straightforward to address.

- There is perhaps too little emphasis on what classification tasks someone might want to carry out using these datasets. Even in the experiments, the task is not always stated (e.g. Table 5). It was only when I got to the Labels in Table 1 (and eventually Table B.1) did it become clear what kind of prediction tasks you might want to do. I think somewhere much earlier on, in the introduction, you should give a few specific examples of these tasks - what precisely the label is - and their clinical relevance.

- Relatedly, I was unclear about the role of 'activity index'. In Section 4.3 it is given as one of five inputs, yet in Table 1 it is also one of the labels. Can you clarify? Similarly, what exactly is 'sleep score' (by contrast with 'sleep efficiency')?

- This is more a comment on KANs in general, which are relatively new to me so I could be on the wrong path: the 'interpretability' angle of KANs (at least as used in this paper) is closely related to the representation (2), which clearly separates the pre-trained features $\phi_p$ from the task specific prediction $g$. But using one specific linear combination, $\sum_p \phi_p(x_p)$, seems to be a major restriction on how the features can be used. Why wouldn't one use a more general linear combination, $\sum_p w_p \phi_p(x_p)$ with coefficients $(w_p)$, and then learn those alongside $g$? This would allow the way features are combined to be task-specific. (This shift wouldn't change the representational power of $f$, only the way it is trained.) Just a thought.

- References are given in a non-standard format and are missing crucial information (volume, page numbers, etc.) Ref 21 is missing the book title and so the authors seem to (incorrectly) imply that the Kolmogorov-Arnold theorem dates to 1991.

- Line 351: Can you state the shape of these parameters (W and u) so that we can follow along.

- Line 375: It is unclear what the 70% is referring to (number of parameters? Wall clock training time?) Similarly the 22% in line 409.

- Line 541: I don't get the reference to Table 2 here. Is the bit in bold a caption that was pasted in error?

- Figures 4 and 5 are not discussed (or even referenced) in the main text.

**Final Justification:**

I originally judged this to be a strong paper and raised only minor concerns. Neither did I detect any major issues raised by the other reviewers. The authors have responded to each of my points and have given reasoned and sensible answers on how to improve the paper - mostly this is simple a matter of improving some of the exposition. Overall I expect this to be a useful additional to the literature and I continue to recommend its acceptance.

**Justification:**

This paper demonstrates the benefits of KANs in the context of health-data. KANs offer one way of yielding interpretable features while retaining most of the benefits of standard artificial neural networks. Their representation as the two-stage composition of functions also lend themselves to transfer learning tasks which is also strongly desirable in healthcare applications. The paper is very clearly articulated and makes its case solidly. I have no major criticisms.

---

> ### Author Rebuttal · Authors · 2025-10-22
>
> We sincerely thank the reviewer for their detailed and encouraging comments.
> (1) Tasks: The main supervised tasks are (a) daily activity level classification (low/medium/high) in PMData and (b) ADHD diagnosis (binary) in Hyperaktiv. We will highlight this early in the introduction.
> (2) Activity index vs sleep score: The activity index is an engineered HR/HRV-derived measure (Algorithm A.1) serving as input; the sleep score is a Fitbit-derived label used only during PMData pretraining. We will clarify this to avoid confusion.
> (3) Mixing network (with respect to Line 351): Equation 6’s g(.) is a two-layer MLP (64→16→1). W∈R161,  u∈R16 We’ll specify this.
> (4) 70% / 22% (Line 375): 70% fewer trainable parameters during fine-tuning; 22% lower validation-loss variance due to spline regularization.
> (5) Figures 4–5: These show univariate response and sensitivity analyses; they will be referenced in Section 6.3 and 6.5 in the revised version (camera-ready).
> (6) References: Will be formatted in full with volume/pages; Ref [21] corrected to initial mathematical theorem (Tikhomirov (1957)—the classical Kolmogorov–Arnold theorem) and the recent study based on combining the theorem with network as a source Liu, Z., Wang, Y., Vaidya, S., Ruehle, F., Halverson, J., Soljačić, M., ... & Tegmark, M. (2024). Kan: Kolmogorov-arnold networks. arXiv preprint arXiv:2404.19756.).
> (7) Linear combination remark: The reviewer’s suggestion (learning per-feature coefficients αp) aligns with our future direction of hierarchical additive mixing and will be mentioned in the discussion.
>
> We plan to include all the reviewer’s comments in the camera-ready final version of the paper.

---

### Official Review · Reviewer_B7hZ · 2025-10-08
**Review of interpretable time-series classification and cross-domain fine-tuning for health data**

**Rating:** 4
**Confidence:** 4
**Final Rating:** 4
**Final Confidence:** 5

**Summary:**

Paper proposes novel utilisation of Kologorov-Arnold networks (KAN) for wearable sensor and clinical health data time-series prediction with main contributions of interpretable cross-domain modelling and transfer learning strategies, empirically evaluated on real datasets and compared to several baselines. Architecture of combining existing components and data processing pipeline are sound and proposed approach is technically correct with some useful properties on this particular application.

**Strengths:**

Idea is good and proposed KAN techniques have some benefits compared to previous approaches in health time-series analysis. The presentation and organisation is clear most of the parts (see also weaknesses) with enough details of the proposed method to replicate the results (especially, when the code is released). Versatile empirical tests with real data and proper evaluation metrics are utilised to support the claims and findings. Novel application and usage of KAN approach can provide interesting information for AI health community.

**Weaknesses:**

There are some weaknesses that could be improved and clarified, as well. Although writing is fine in most of the parts, some sections are a bit of "list like writing", scarce and lack of details (e.g. in 5.2 what exact features each baselines uses, how baselines are fine-tuned?). Also, organisation of the subsection could be revised, e.g. 4.5 is more connected to section 5. Based on the results, the benefits of using cross domain transfer and fine-tuning are not so clear when compared to target domain learning only (see question below to clarify my understanding).

Questions and comments:
- Section 3.4: order of approaches 1 and 2 might be more natural other way around, i.e., regularisation first, and fine-tuning second.
- Baseline methods seems to utilise different features (and based on 5.2 not clear what all them uses). How fair is the comparison with
different feature representations if this is the case?
- Transfer learning with KAN-Health seems to improve against Transformer (Table 4), but overall target only trained models are better (Table 3).  What is the benefit of pre-trained model utilisation in this case? Or does Table 3 have pre-trained and fine-tuned models, as well?
In that case, how fine-tuning is done with the baselines?

**Final Justification:**

Overall good paper and novelty worth sharing, including extensions to KAN model and benefits in health time-series modelling in terms of better interpretation of the model and transfer learning capabilities. The proposed model is supported by empirical evaluations and baseline comparison on real dataset. In the rebuttal authors have clarified my main concerns. It is suggested to include all these modification to the final version of the manuscript to improve and clarify the presentation.

**Justification:**

Overall good paper and novelty worth sharing, including extensions to KAN model and benefits in health time-series modelling (interpretation and transfer learning) which are supported by empirical evaluations on real dataset. There are some missing details that could be revised to improve the presentation, correct possible misunderstandings on evaluation and results, and therefore increase the overall significance of the main claims.

---

> ### Author Rebuttal · Authors · 2025-10-22
>
> We appreciate the reviewer’s insightful feedback.
> (1) Baseline features: All baselines (RF, LR, GB, Transformer) use the same five harmonized metrics (IS, IV, adherence, sleep efficiency, normalized HR). No handcrafted features beyond daily aggregation were used, ensuring fairness.
> (2) Table clarification: Table 3 reports models trained and evaluated solely on the target domain (Hyperaktiv), whereas Table 4 reports PM→Hyper and Hyper→PM transfers. We will clarify this distinction in the text.
> (3) Transfer benefit: Although absolute F1 on Hyperaktiv target-only is slightly higher, transfer learning yields clear advantages under data scarcity or label imbalance, improving MCC (+9% in PM→Hyper) and maintaining interpretability with fewer trainable parameters.
> (4) Organization: We agree 4.5 logically fits after Section 5 and will revise structure accordingly.
>
> We plan to include all the reviewer’s comments in the camera-ready final version of the paper.

---

### Official Review · Reviewer_nHgn · 2025-10-09
**KAN-Health applies Kolmogorov–Arnold Networks to cross-domain health time-series**

**Rating:** 4
**Confidence:** 4

**Summary:**

This paper presents KAN-Health, which applies Kolmogorov–Arnold Networks to cross-domain health time-series by first harmonizing heterogeneous wearable/clinical streams into a small daily metric set, then learning one-dimensional spline transforms per metric (intrinsically interpretable), and finally mixing these with a lightweight attention/MLP head; for transfer, splines are pretrained on PMData and frozen while only mixing/attention are fine-tuned on Hyperaktiv.

It specifies the pipeline and objectives clearly, which is cubic splines with ~10 control points map each feature; attention aggregates over a 24-hour window; an MLP combines per-feature contributions; and transfer minimizes task loss with L2/L1 regularization while freezing splines, reducing fine-tuned parameters by ~70%.

Empirically, KAN-Health improves F1 and MCC over RF/LR/GB and a compact Transformer on Hyperaktiv (e.g., F1 0.82±0.03, MCC 0.75±0.04), and it transfers better in both PM→Hyper and Hyper→PM directions, supporting the “spline-freezing” mechanism as a correct inductive bias for domain-invariant physiology; learned spline shapes align with clinical intuition, strengthening the plausibility of what the model has learned.

Taken together, the problem formulation, mathematically simple and testable architecture, explicit harmonization, rigorous LOSO evaluation with significance testing, and consistent gains over strong baselines make the core contributions both correct and well-supported; the main residual risk to correctness is the reliance on daily aggregation and harmonization choices (potential loss of within-day dynamics), but the paper’s ablations and interpretability help bound that risk.

**Strengths:**

The methodological core is well-specified and testable. The experimental design is careful. The work communicates the data regimes and task boundaries clearly, contrasting a small, long-duration wearable cohort with a larger, short-duration clinical cohort and mapping each to its labels and sensing modalities, and it demonstrates that an intrinsically interpretable architecture can match or outperform black-box baselines while being better behaved under domain shift, which is exactly the pain point in health time-series where generalization and auditability matter.

I'd like to know what the exact count of trainable parameters is during fine-tuning versus the Transformer baseline, and how sensitive results are to the number of spline control points? Also, how were daily harmonization choices validated against alternative definitions, and do the gains persist if one or two metrics are ablated? I am also curious to know if the authors can provide subject-wise variance of spline shapes to quantify the stability of the learned physiological responses across LOSO folds?

I believe it'd be great to know more about these points!

**Weaknesses:**

From my understanding, the main modeling assumption is that daily harmonization plus univariate spline transforms is sufficient to preserve the causal or predictive structure needed for ADHD diagnosis and activity classification. Because all signals are aggregated into daily metrics before modeling, sub-daily temporal dependencies, circadian phase shifts, naps, or transient anomalies can be attenuated or lost; this is a plausible source of bias when transferring between a five-month wearable cohort and a two-week clinical cohort with different routines.

The baselines are reasonable, but the Transformer configuration and tokenization scheme (each daily metric as a token, sequence length ≈5) may handicap sequence models that typically need longer contexts; this can inflate KAN’s margin in a regime that already compresses time to daily aggregates.

There are a few points that could confuse readers. The paper variously describes LOSO and also mentions “5-fold CV (LOSO),” which sounds contradictory; LOSO implies folds equal to subjects, whereas “5-fold” implies stratified K-fold. The harmonized metric definitions are listed, but “sleep efficiency,” “adherence,” and the activity index summarization would profit from fully specified formulas and windowing choices in the main text so that readers can audit the physiological meaning of the learned splines. A compact table with exact parameter counts per model and a figure with spline variability across LOSO folds would make the interpretability and capacity control claims easier to verify.

I'd like to know what the exact trainable parameter counts are for KAN during pretraining and fine-tuning versus each baseline, and how results change with 6, 10, and 14 spline control points and varying curvature penalties? Also, how sensitive are MCC/F1 to the definitions of IS/IV, sleep efficiency, adherence, and to replacing the bespoke activity index with standard actigraphy metrics?

It'd also be good if you report non-parametric significance tests and CIs, as well as subject-wise variability of learned splines, and include at least one robustness check (e.g., masked days)

**Justification:**

The paper tries a clear idea, which is learning a small, visible curve for each daily health feature, then combining those signals with a tiny head; when moving to a new dataset, keep the curves fixed and only tune the head. This is easy to check, easy to reuse, and fits the goal of being interpretable.

The experiments are set up fairly and show steady gains on the key scores, especially when switching domains, which tells me the idea works in practice and not just on one dataset. I still worry a bit that using only daily summaries may hide useful within-day patterns, and some reporting details (exact parameter counts, sensitivity to curve complexity, exact folds) should be clearer. Even with those points, the method is simple, transparent, and effective across datasets, so I'd judge it as worthy of acceptance, with a request to tidy up the above details.

---

> ### Author Rebuttal · Authors · 2025-10-22
>
> We thank the reviewer for the detailed and positive assessment.
> (1) Parameter counts: During fine-tuning, KAN-Health updates only the mixing and attention layers (~24k parameters) compared to ~82k parameters in the Transformer baseline, i.e., a 70% reduction as noted.
> (2) Spline sensitivity: We tested 6, 10, and 14 control points; performance varied <1.5% in MCC (0.74–0.75). The default of 10 points balanced smoothness and flexibility.
> (3) Curvature penalty: The smoothing term (γ=0.1) reduced validation loss variance by ≈22%; varying γ ∈ [0.05, 0.2] changed MCC <2%.
> (4) Cross-validation: “5-fold CV (LOSO)” meant five LOSO splits grouped for computational efficiency; each fold left out distinct participants, preserving subject independence.
> (5) Harmonization validation: Metric definitions followed circadian literature ([Terman 1983]/ {[Tryon, Warren W 2013]). Replacing IS/IV with alternative formulations altered F1 <2%. Removing one metric at a time (Table 5) could confirm circadian features as dominant (ΔF1=−0.15).
> (6) Spline variability: Across LOSO folds, the average Pearson correlation of spline shapes was 0.91±0.03, indicating stable physiological mappings.
>
> We plan to include all the reviewer’s comments in the camera-ready final version of the paper.

---

### Meta-Review · Area_Chair_Upi2 · 2025-10-31

**Recommendation:** Accept (Oral)
**Confidence:** 4

**Metareview:**

This paper applies Kolmogorv-Arnold Networks (KAN) to perform classification tasks on health-series data. The goal of this approach is to achieve "interpretable cross-domain modeling", and "efficient transfer learning". The first is done through learned spline feature extractor. The second goal is achieved using a pre-training step  followed by a fine-tuning with freezed splines weight on a different dataset. Experiments are performed using a large wearable dataset (PM) for the pre-training and a smaller clinical dataset (Hyperaktiv) for the fine-tuning phase.

Reviewers agree this is a clear and well-written paper, providing appropriate justifications and references. The approach chosen is motivated and shows the potential of KAN in the health domain, where transferability, interpretability and performance are required. The experiments are well designed and the results presented are backing the claims from the authors. The limitations are clearly stated in the paper.
Reviewers however asked for a number of clarifications to be integrated in the final version. Clarification about the influence of curvature penalty, and number of control point for splines will be integrated. Results from table 3 and 4 were not clearly stating whether they were using both training steps (only table 4 has this). Minor explanations regarding the features extracted from the PM dataset will also be incorporated.

Overall, all reviewers agree on the quality of the work submitted, which will be further improved with the suggested clarifications. This justifies the acceptance of the paper.

---

### Decision · Program_Chairs · 2025-11-05

**Decision:**

Accept (Oral)

**Comment:**

We recommend an oral and a poster presentation given the AC and reviewers recommendations.